# Barrier-on-a-Chip with a Modular Architecture and Integrated Sensors for Real-Time Measurement of Biological Barrier Function

**DOI:** 10.3390/mi12070816

**Published:** 2021-07-12

**Authors:** Patrícia Zoio, Sara Lopes-Ventura, Abel Oliva

**Affiliations:** 1Instituto de Tecnologia Química e Biológica (ITQB), Universidade Nova de Lisboa, Avenida da República, Estação Agronómica Nacional, 2780-157 Oeiras, Portugal; patricia.zoio@itqb.unl.pt (P.Z.); sara.ventura@itqb.unl.pt (S.L.-V.); 2Instituto de Biologia Experimental e Tecnológica (IBET), 2781-901 Oeiras, Portugal

**Keywords:** organ-on-chip, transepithelial electrical resistance, barrier-on-chip, skin-on-chip, barrier tissue, reconstructed skin model

## Abstract

Biological barriers are essential for the maintenance of organ homeostasis and their dysfunction is responsible for many prevalent diseases. Advanced in vitro models of biological barriers have been developed through the combination of 3D cell culture techniques and organ-on-chip (OoC) technology. However, real-time monitoring of tissue function inside the OoC devices has been challenging, with most approaches relying on off-chip analysis and imaging techniques. In this study, we designed and fabricated a low-cost barrier-on-chip (BoC) device with integrated electrodes for the development and real-time monitoring of biological barriers. The integrated electrodes were used to measure transepithelial electrical resistance (TEER) during tissue culture, thereby quantitatively evaluating tissue barrier function. A finite element analysis was performed to study the sensitivity of the integrated electrodes and to compare them with conventional systems. As proof-of-concept, a full-thickness human skin model (FTSm) was grown on the developed BoC, and TEER was measured on-chip during the culture. After 14 days of culture, the barrier tissue was challenged with a benchmark irritant and its impact was evaluated on-chip through TEER measurements. The developed BoC with an integrated sensing capability represents a promising tool for real-time assessment of barrier function in the context of drug testing and disease modelling.

## 1. Introduction

Biological barriers are crucial for the integrity and proper function of various organs, including the skin, brain, eye and blood vessels [1]. Disruption and dysfunction of barrier-forming tissues are an integral part of the pathophysiology of many disorders (skin barrier in psoriasis [2], blood–brain barrier in multiple sclerosis [3], blood retinal barrier in macular degeneration [4], endothelial barrier in neurodegenerative disorders and ischemic stroke [5,6]). An understanding of barrier function is essential when studying the causes and mechanisms of disease as well as when developing novel drugs and treatments.

The importance of biological barriers pressured the development of in vitro methods to recreate and characterize the tissue barrier function. Permeable cell culture supports (e.g., Transwell^®^) are often used to generate in vitro organ-tissue models [7,8,9]. However, these conventional culture systems do not provide a dynamic physicochemical microenvironment which plays an important role in maintaining the barrier function of tissues. Organ-on-chip (OoC) systems show great promise in overcoming these challenges by including key physical and biochemical parameters [10,11,12]. To maintain a suitable environment for tissue growth inside these systems it is necessary to monitor physiological parameters in real-time. Currently, monitoring tissue function inside the OoC mainly depends on endpoint assessment techniques. In particular, for 3D cell culture using scaffolds or membranes, this monitoring cannot be performed using conventional microscopy techniques due to difficulties with light penetration and scattering effects [13].

Transepithelial/transendothelial electrical resistance (TEER) has become a widely accepted method to evaluate the tissue barrier function in in vitro systems [14]. This method is a label-free alternative to the widely used conventional assays based on the transport of tracer compounds. TEER enables non-destructive, real-time quantification of the barrier function. Conventional TEER measurements are performed by introducing commercially available chopstick-type electrodes at both sides of a cellular barrier. The resistance of the path between the electrodes is determined by applying an alternating current (AC) signal at low frequency (typically 12.5 Hz). However, TEER readings using handheld chopstick electrodes have low reproducibility due to variations in depth and/or the angle of immersion. Moreover, chopstick-type electrodes cannot deliver a uniform current density when large tissue culture inserts are used (>12 mm in diameter) [15]. Alternatively, EndOhm chambers contain fixed concentric electrodes, symmetrical positioned at both sides of the cellular barrier. This results in a more uniform current density when compared to chopstick electrodes and in a higher reproducibility [14]. However, cell culture inserts need to be manually transferred from the wells to the chambers which can increase the probability of tissue damage and contamination.

Just as in conventional systems, the TEER of a cellular barrier inside an OoC device can be measured by introducing electrodes on both sides of the cellular barrier. This has been achieved by inserting electrodes on the chip inlets/outlets [16,17,18], in a process similar to the introduction of the chopstick electrodes. Alternatively, the OoC with integrated TEER electrodes has been fabricated for development and monitoring of the blood–brain barrier [19,20], gastrointestinal tract [21,22] and airway epithelium [21]. The integration of the electrodes during the fabrication reduces the noise generated by electrode motion. However, most of the devices described in the literature require expensive processes and advanced microfabrication techniques for the development of the OoC and for the patterning of the TEER electrodes on the substrates of the channels, often requiring cleanroom access [23]. Currently, these costs of fabrication and the use of materials and strategies unsuitable for mass production are some of the major barriers delaying the transfer of the OoC to the industry [24]. Furthermore, the process of PDMS bonding makes it difficult to integrate scaffolds for 3D cell culture which are typically an order of magnitude thicker than membranes [25].

Here, we have developed a low-cost chip with a modular architecture and integrated custom-made tetrapolar electrodes for the development and monitoring of a biological barrier-on-chip (BoC). The fabrication of this platform does not require plasma bonding and it is cleanroom-free. The geometry of the custom-made electrodes integrated on the BoC is similar to the electrode arrangement on the EndOhm chamber, with a symmetric and concentric placement on both sides of the cellular barrier. Taking into account works from Odijik et al. [26] and Yeste et al. [15], highlighting possible sources of error when integrating tetrapolar electrodes on OoC, we use finite element analysis (FEA) to determine the sensitivity distribution within the cell culture chamber. These simulations are performed for electrodes of different diameters and compared with FEA of chopstick electrodes for measuring the TEER in permeable inserts of a smaller (6.5 mm diameter) and larger culture area (12 mm diameter).

As proof-of-concept, we show the applicability of our device by developing a full-thickness human skin model (FTSm) inside the system. This is achieved by co-culturing primary human fibroblasts and primary human keratinocytes on an inert polymeric scaffold inserted between two perfusion channels. Various skin-on-a-chip (SoC) devices have been fabricated in an attempt of reproducing a skin model with a better barrier function by including fluidic forces [18,27,28,29,30]. However, SoC devices face multiple technical challenges, including the tendency of the hydrogel-based materials (e.g., collagen) to contract over time leading to tissue disruption and the difficulty in recreating the complex 3D architecture of the epidermis [31]. To our knowledge, until now, there is no description of a functional OoC combining 3D skin production with real-time TEER measurement. Here, we measured the TEER in real-time, following skin differentiation and resultant barrier formation. Histological analysis was performed on tissues representative of different stages of differentiation. TEER-on-chip measurements were performed to test the impact of a benchmark irritant on the tissue barrier function.

The presented platform is applicable to any biological barrier requiring an apical and basolateral chamber to mimic tissue–tissue, tissue–liquid and tissue–air interfaces. Moreover, the possibility of integrating other structures besides membranes (e.g., polymeric scaffolds) opens the door for reproducing tissues with an increased level of complexity.

## 2. Materials and Methods

### 2.1. Simulation Model

FEA was performed using the AC/DC module from the commercial software COMSOL Multiphysics v.5.3 (COMSOL Inc., Burlington, MA, USA) to map the sensitivity field (S) distribution for different tetrapolar electrode configurations placed within different chamber geometries. Figure 1 provides details of the different electrode and chamber configurations used for FEA: (1) BoC device with integrated custom-made electrodes with different dimensions (2 mm and 4 mm outer diameter) (Figure 1a); (2) conventional permeable insert model with chopstick electrodes and cell culture area with different dimensions (6.5 mm diameter and 12 mm diameter) (Figure 1b).

In more detail, the BoC model comprises two chambers and a small volume at the centre representing the 3D tissue. Concentric electrodes representing the current carrying (CC) and voltage sensing (VS) electrodes were placed in upper and lower surfaces on the top and bottom chambers. Two different electrode sizes were simulated: CC electrodes 0.8 mm + VS electrodes 2 mm diameter (BoC A) and CC electrodes 2 mm + VS electrodes 4 mm diameter (BoC B). The 3D geometry of the BoC and the complete mesh can be seen in Figure 2a. The tetrahedral mesh consisted of approximately 7 × 10^4^ domain elements, 2 × 10^4^ boundary elements, and 1 × 10^3^ edge elements.

The insert model is a 3D representation of a conventional permeable support for cell culture and chopstick electrodes. The model was created for 12 mm (Insert A) and 6.5 mm (Insert B) diameter inserts. These diameters are widely used to generate biological barriers and recommended for normalizing TEER measurements performed with chopstick electrodes. The 3D geometry of this model and the complete mesh can be seen in Figure 2b. Complete mesh consists of approximately 2 × 10^5^ domain elements, 2 × 10^4^ boundary elements, and 1 × 10^3^ edge elements.

The electrical current (ec) interface of the AC/DC module was used to calculate the electric fields and current density distributions associated with the analysed electrode configurations. For all configurations, a constant DC current of 1A was injected through the current carrying (CC) electrodes and the pick-up, voltage-sensing (VS) electrodes were set as floating potential and zero current. The other outer boundaries were defined as electrical insulations. For all models, the “extra fine” mesh (under COMSOL calibration for general physics) was chosen, automatically assigning mesh sizes.

More details about the FEA parameters are provided as Appendix A. To reduce the computational complexity of the simulations, time-independent DC simulations were performed. This is a valid approximation to the experimental apparatus since a low frequency (12.5 Hz) AC current was used.

### 2.2. Sensitivity Evaluation

The sensitivity of a small volume within the measured biomaterial is a measure of how much the resistivity of this volume contributes the total measured impedance, provided that the electrical properties are uniform throughout the material. For calculation of sensitivity, the reciprocity theorem was used [32]. The sensitivity S of each zone can be calculated as reported by Grimnes and Martinsen [33]:(1)$$S=\frac{J_{CC}\;\cdot \;J_{VS}}{I^{2}}$$
where $J_{CC}$ is the current density field that results by using CC electrodes for current injection, $J_{VS}$ is the current density field produced by reversing the setup and using the VS electrodes for current injection, and I is the magnitude of the applied current. The sensitivity of a tetrapolar system was computed using FEA, by:Applying a constant current I between the two CC electrodes and computing the current density field $J_{CC}$ in each small volume in the material as a result of this current.The CC and VS electrodes are exchanged, i.e., the same current I is injected between the voltage VS electrodes and again the resulting current density $J_{VS}$ is computed in each small volume element.The vector dot product between $J_{CC}$ and $J_{VS}$ in each volume element, divided by the current squared, is the sensitivity of the volume element.

Sensitivity can be positive, negative or zero depending on the angle between $J_{CC}$ and $J_{VS}$. Positive sensitivity means that an increase in electrical resistivity of a volume results in a corresponding increase in the measured electrical resistance. Conversely, negative sensitivity results in a decrease in the measured electrical resistance for the same increase in resistivity. In areas of zero sensitivity, a change in electrical resistivity does not notably affect the impedance. Since the sensitivity is normalised to the current, this distribution is independent of the current used.

Simulations were conducted with parametric sweeps of the TEER (from 10^1^ to 10^4^ Ω·cm^2^) spanning the range of the values reported in the literature for barrier tissues as well as simulating the growth and development of these tissues [24]. The sensitivity distribution along the cell barrier surface was simulated and compared between the different models.

### 2.3. Fabrication and Assembly of Barrier-on-a-Chip (BoC) Device

The fabricated device includes two polystyrene compartments (apical and basal) and a central layer for tissue culture. The central layer is composed of polydimethylsiloxane (PDMS) and a thin layer of polymerase chain reaction (PCR) tape to seal a membrane or scaffold at the centre. The basal compartment includes a fluidic channel and inlet/outlet channels for perfusion with culture media or flow-through of a receiver solution in an in vitro test. The apical compartment presents the same fluidic structure and can be used for perfusion of culture medium (liquid–liquid interface) or ventilation (air–liquid interface). An exploded view of the BoC can be seen in Figure 3. The design includes integrated magnets to correctly seal the cell culture layer between the apical and basal compartments.

The fluidic compartments and the moulds used to produce the cell culture layer were designed through computer-aided design (CAD) using AutoCAD 2017 (Waltham, MA, USA) and fabricated using computer numerically controlled (CNC) machining and laser machining (Xometry Europe GmbH, Ottobrunn, DE, Germany). The apical and basal chamber were constructed in polystyrene with the following dimensions: 31.5 mm in length, 31.5 mm in width, 4 mm in heigh and a central cavity with 21 mm in length, 21 mm in width and 1 mm in height. The cavities were included on the fluidic chambers to accept the PDMS layer for skin culture. Each of the fabricated layers contained 4 neodymium magnets with 4 mm diameter and 3 mm height with a force of 3N each. The magnets were placed with polar opposites facing each other on the basal and apical layer. Both apical and basal compartments included a central circular opening housing the electrodes for TEER measurements. For the access ports, mini female luers (microfluidic ChipShop, Jena, DE, Germany) were glued to the inlets and outlets into the upper and lower polystyrene housings with the channel features oriented to the interior of the device.

To obtain central layers for cell culture that could fit in the polystyrene cavities, moulds were fabricated using CNC machining and laser cutting. The base of the mould was fabricated in PMMA and had 21 mm in length and 21 mm in width and included a central cylinder with 5 mm diameter and 2 mm height. The lateral and top pieces of the mould were fabricated with laser cutting using PMMA with 2 mm in thickness. The mould cavity was filled with degassed PDMS prepolymer mixed with curing agent (SYLGARD 184, Dow Corning, Midland, MI, USA) at a 10:1 (*w*/*w*) ratio. The assembled moulds were clamped and placed in an oven for 12 h at 65 °C. The resulting PDMS layer was removed from the mould with the help of a razor blade. The polystyrene scaffolds (Alvetex^®^, REPROCELL Europe Ltd., Glasgow, UK) were cut with a 6 mm diameter puncher and sandwiched between the PDMS insert layer and PCR tape. To better seal the PDMS insert against the polystyrene housing and avoid leakage, the PCR tape was covered with a thin PDMS layer. This geometry ensured that the communication between channels only occurred though the area at the centre where the cells were seeded. The modular approach of the design makes it possible to handle the cell culture layer independently from the fluidic network.

The electrodes integrated at the apical and basal layers were both made of a 0.8 mm diameter Ag/AgCl-sintered pellet electrode (NEUROSPEC AG, Stans, Switzerland), an electrical insulating polyether ether ketone (PEEK) tubing and silver tubing with 2 mm OD. They were concentrically assembled and installed on both perfusion chambers (Figure 3b). Liquid PDMS pre-polymer mixed with curing agent was applied to fill gaps and cured at 50 °C for 24 h to ensure leakage-free installation. The BoC devices were sterilized using UVC radiation before each experiment.

### 2.4. Primary Cells and Cell Maintenance

Primary human foreskin-derived dermal fibroblasts (HDFn, CELLnTEC, Bern, Switzerland) were maintained in Fibroblast Growth medium (FGM) composed of Iscove’s Modified Dulbecco’s Medium (IMDM Gibco^®^, Waltham, MA, USA) supplemented with 10% foetal bovine serum (FBS, Gibco^®^, Waltham, MA, USA), at 37 °C in a 5% CO_2_ humidifier, following supplier’s instructions. HDFs were used for up to 10 passages and subcultured at a 90% confluency.

Primary human epidermal keratinocytes isolated from neonatal foreskin (HEKn, Gibco^®^, Waltham, MA, USA) were maintained in keratinocyte growth medium (KGM) composed of EpiLife medium (Gibco^®^, Waltham, MA, USA), supplemented with 0.06 mM calcium and keratinocyte growth factor (HKGS, Gibco^®^, Waltham, MA, USA), at 37 °C in a 5% CO_2_ humidifier incubator. The KGM medium was changed every other day until the cells reached 50% confluency. At this point the medium was changed every day. A low passage (4–5) and actively proliferating HEKns were used for successful 3D culture. All cultures were routinely monitored for contaminations.

### 2.5. Generation of Full Thickness Equivalents on the BoC

The process of developing fully human FTSm can be divided in three main steps: the development of a mature dermis, the culture of HEKns on top of the dermis under submerged conditions, and the culture of the epidermis at the air–liquid interface (ALI) until a fully differentiated skin is produced. The protocol for generating an FTSm using a polystyrene scaffold and a fibroblast-derived matrix was adapted from [34].

Briefly, dermal models were generated by seeding HDFns ($3.0\times 10^{5}$ cells) onto Alvetex^®^ scaffolds (REPROCELL Europe Ltd., Glasgow, UK) and incubating at 37 °C in a 5% CO_2_ humidified incubator in FGM medium supplemented with 100 µg/mL L-ascorbic acid 2-phosphate (Sigma-Aldrich, St. Louis, MO, USA). Dermal equivalents were maintained for 12 days on a 6-well plate under static conditions.

FTSm were generated by seeding HEKns onto the dermal equivalents and maintaining the tissue construct on the BoC under submerged conditions for 2 days and ALI for up to 14 days. For this, HEKns were harvested by trypsinization and 30 μL of cell suspension ($5.0\times 10^{6}\;\mathrm{cells}/\mathrm{mL}$) was seeded on top of the dermal equivalent in KGM containing high calcium concentration (1.5 mM). The tissue construct was incubated for 3 h for cell adhesion at 37 °C in a 5% CO_2_ incubator. After the incubation period, the PDMS layer was inserted on the BoC, between the apical and basal chambers, and parallel perfusion was established. The inlet ports on the top and lower compartments were connected to the tubing mounted on the peristaltic pump and connected to a media reservoir with supplemented KGM medium. The medium was pumped from the reservoir through the inlets of the apical and basal chambers and left the BoC through the apical and basal outlets. With this configuration, the FTSm in culture chamber of the device was double-sided perfused at a flow rate of 1.5 μL/min.

After 48 h, the models were raised to ALI and cultured in KGM containing 1.5 mM calcium, supplemented with 10 ng/mL KGF and 50 µg/mL of L-ascorbic acid 2-phosphate and maintained up to a further 14 days. For this, the medium was pumped only through the basal chamber at a flow rate of 1.5 μL/min and air was pumped through the apical chamber to allow the formation of a differentiated epidermis.

### 2.6. TEER Measurements

TEER was measured with the EVOM voltohmmeter (WPI Europe, Friedberg, Germany), using the tetrapolar method. The voltohmmeter supplies an AC square-wave current of ±10 μA at 12.5 Hz through the outer CC electrodes and measures the voltage drop across the inner VS electrodes as shown schematically in Figure 3c. This technique was used to measure TEER using conventional chopstick electrodes and concentric electrodes integrated in the BoC device. For both cases, each electrode pair contains a silver/silver-chloride pellet for measuring voltage and a silver electrode for passing current. TEER values were recorded during culture time from the last day under submerged conditions (day 0) until day 14 at ALI. For this, the medium was pumped through both the basal and apical chambers to bridge an electrical connection between top and bottom sensors. After a 10 min TEER reading for each sample, the ALI was resumed. At the removal of the medium from the apical chamber, the TEER becomes immeasurable again due to the removal of the electrical bridge. These measurements were performed for 3 skin batches at each time point. TEER was also measured for a blank scaffold and subtracted from each recording.

The applicability of the TEER sensors for toxicological applications was accessed by delivering cell medium with 0.2% sodium dodecyl sulphate (SDS) to the basal side of the skin. TEER measurements were recorded at day 14 of ALI by pumping phosphate buffered saline (PBS) into the apical chamber and medium with SDS to the basal chamber to bridge an electrical connection between sensors. Perfusion with SDS medium continued for 3 h with TEER being measured every 15 min.

To better validate the custom-made tetrapolar electrodes, a comparison on the resistance readings was determined between the BoC electrode system and the traditional chopstick electrodes. Chopstick electrodes are the conventional method to perform TEER readings on permeable cell culture inserts and present approximate uniform current distributions for small diameters and high TEER values. Since it is not possible to directly introduce the chopstick electrodes on the BoC device, a custom calibration chamber was designed and fabricated (Appendix A). The chamber was fabricated using precision milling and was designed to accommodate the PDMS insert layer while allowing for the same placement of the chopstick electrodes as in the conventional permeable inserts for cell culture.

### 2.7. Barrier Function Analysis with Passive Dye

The barrier function of the skin models was further assessed by the penetration of the passive dye lucifer yellow (Sigma-Aldrich) in combination with treatment with the detergent SDS. Solution of SDS 0.2% *w/v* in PBS was perfused at the apical chamber for 0, 1 and 3 h. After perfusion with SDS, the models were washed three times with PBS. An amount of 1 mg/mL lucifer yellow was topically applied to the models for 30 min, followed by three washes with PBS. The tissues were fixed with neutral buffered 10% formalin solution (Sigma-Aldrich) and embedded in paraffin. Tissue sections were rehydrated and nuclei were stained with 1 μg/mL of 4,6-diamidino-2-phenylindole (DAPI, Invitrogen, Waltham, MA, USA) and slides were mounted with VECTASHIELD (Vector Laboratories, Burlingame, CA, USA). Images were obtained using the Nikon Eclipse TE2000-S fluorescence microscope (Nikon instruments, Melville, NY, USA) and analysed with the ImageJ software.

### 2.8. Histochemistry Analysis

The modular architecture and reversible sealing made it possible to easily open the platform and remove the tissue for analysis. The reconstructed tissues were fixed immediately after being taken out of chip in 10% neutral buffered formalin (Sigma). The samples were embedded in paraffin to allow for transverse sectioning. Paraffin-embedded sections (5 µm-thick) were de-paraffined and re-hydrated for morphological evaluation by staining with haematoxylin and eosin (H&E) through standard methods, as described in [35].

## 3. Results

### 3.1. Sensitivity Distribution

In this study, the software COMSOL Multiphysics was used to model the complex BoC and insert geometries and perform FEA. The sensitivity field along the tissue barrier area was evaluated using Equation (1). This was used to determine the contribution of each zone of the tissue to the measured resistance for the different electrode configurations and chamber geometries.

Figure 4 shows the planar sensitivity at the tissue culture plane and the normalized sensitivity (*S*_normal_) profile along the line AA’ when TEER is measured using a BoC device with integrated concentric electrodes (Figure 4a) covering ~15% of total culture area (BoC A) or covering ~65% of total culture area (BoC B) and when TEER was measured using chopstick electrodes and an insert with conventional sizes (Figure 4b): 12 mm diameter (insert A) and 6.5 mm diameter insert (insert B).

For all the setups, the sensitivity values were positive in the entire cell culture volume. Zones of negative sensitive can be seen close to the electrodes (Appendix A). The sensitivity was non-uniformly distributed with the highest values in the regions close to the electrodes and lowest values (normalized to 0%) in the regions further away from the electrodes. As expected, for both BoC and insert models, regions closer to the electrodes contributed more to the measured resistance. Moving away from the electrodes, the sensitivity rapidly decreased. In the BoC device this can be seen by the central areas of increased sensitivity below the electrodes, for both configurations (Figure 4a). In the insert models, sensitivity peaks can be seen in the periphery below the electrodes, representing a higher contribution of these regions to the resistance measurement (Figure 4b). For these configurations, sensitivity valleys can also be observed in the areas in the middle of a pair of electrodes, especially for the model insert B.

A range of the most common TEER values measured on barrier tissues (10^1^ to 10^4^ Ω·cm^2^) was used to study the dependency of this parameter on the sensitivity field. For all the models, there was a dependency of the TEER on the sensitivity, with higher TEER values resulting in a decrease in the peak sensitivity. Thus, resulting in a more uniform sensitivity field.

Concerning the BoC models (Figure 4a), it could be theorized that larger electrodes, covering more tissue area, could result in a more uniform sensitivity. However, this was only verified for very low TEER values (10^1^ Ω·cm^2^). For most TEER values (≥10^2^ Ω·cm^2^), the smaller electrode assembly (BoC A) resulted in a more uniform sensitivity than the larger electrode assembly (BoC B). In particular, for BoC A, TEER values of 10^2^ Ω·cm^2^ resulted in a maximum sensitivity deviation of 17% at the centre of the tissue when comparing with the periphery. TEER values 10^3^ Ω·cm^2^ resulted in a maximum sensitivity deviation of 10% at the centre when comparing with the periphery.

For most TEER values, BoC B resulted in larger differences between the zones of the culture area. For BoC B and TEER of 10^1^ Ω·cm^2^, the central area had a sensitivity 70% higher compared with the periphery. For TEER values ≥10^2^ Ω·cm^2^, the central area had a sensitivity 42% higher compared with the periphery. For the model BoC A, no sensitivity differences were seen for TEER values greater than 10^3^ Ω·cm^2^ and for the model BoC B, no sensitivity differences were seen for TEER values greater than 10^2^ Ω·cm^2^.

Another feature that has an impact on the sensitivity is the geometry and dimensions of the chambers. The sensitivity field in the insert model was less uniform as its diameter increased, because larger inserts have zones further away from the chopstick electrodes that contribute little to the measurement (Figure 4b). For example, for the model insert A and TEER values between 10^1^ and 10^3^ Ω·cm^2^, the sensitivity field below the electrodes contributed 10^3^% more to the measured resistance compared to the regions further away from the electrodes. Thus, resistance measurements performed with this configuration are only representative of small zone of the total culture area, revealing an error when multiplying the measured values by the total area. The most uniform sensitivity distribution was found for the insert B model when measuring tissue in the range of 10^3^ Ω·cm^2^, with maximum sensitivity deviations of 15%. For the calibration chamber, the sensitivity distribution was less homogenous due to the narrowness of the chamber obstructing the electrical current to flow through the cell barrier in areas away from the electrodes (Appendix A).

### 3.2. Barrier-on-Chip with Integrated Sensors: Design and Operation

The fabricated BoC consists of two polystyrene layers that firmly press the central PDMS layer containing a 200 μm thick porous scaffold through magnetic latching or by using screws. Screws were used for long-term studies (tissue formation) and magnetic latching for shorter experiments (irritant exposure). The fabrication of the device was completely cleanroom-free and excluded the use of plasma activation. The BoC correctly sealed the tissue between the basal and apical sides with no signs of fluid leakage. Two isolated chambers were obtained, compatible with the establishment of ALI or liquid–liquid interface. Moreover, since the basal and apical chambers were electrically isolated from one another, TEER measurements could be successfully performed. For this, electrodes were inserted on both chambers and fixed with PDMS, securing the position of the electrodes; therefore, eliminating measurement errors due to variation in electrode placement, which can occur when inserting wires manually into the in/outlets.

Geometrical dimensions of the fabricated BoC slightly differed from those in model BoC A (Figure 1a) and resulted in an asymmetric placement of the cell culture insert between the electrodes. The sensitivity distribution for this electrode configuration was calculated as in the previous simulation study (Appendix A). The asymmetric placement of the cell culture resulted in a sensitivity approximately 20% higher at the centre when comparing with the periphery.

The electrode system incorporated in the BoC was compared to the conventional chopstick electrodes. For this, a calibration chamber was designed and fabricated using precision milling. This calibration chamber could accommodate the insert layer from the BoC and was compatible with the use of chopstick electrodes to perform TEER measurements. Each measurement was performed during 10 min and the variation during this period was recorded. The TEER data obtained with the two devices showed a correlation slope of approximately 0.86 (Appendix A). The values from the chopstick system were lower than the values obtained from the electrodes incorporated in BoC. However, the two systems presented a linear correlation. Compared with the BoC electrodes, measurements performed by the chopstick electrodes resulted in significant variability, mainly due to gesture errors using these electrodes. The electrode position could not be defined precisely and this caused the fluctuations. TEER values obtained using chopstick electrodes presented high variability during the 10-min period reaching 37% variability in the TEER reading (Appendix A). This was also probably due to an incorrect immersion of the chopsticks using the calibration chamber. TEER measurements obtained using the electrodes integrated in BoC device were very stable and presented a maximum deviation of 5% the TEER value for each measurement.

### 3.3. Reconstructed Skin-on-a-Chip Recapitulating In Vivo Skin

A fully human dermis was developed off chip by seeding HDFns on a porous and inert scaffold. For 12 days, these cells were stimulated to produce extracellular-derived matrix (ECM) components which built up and accumulated inside the scaffold. The resultant structure can be seen in Figure 5 (top left), with a homogeneous distribution of HDFns and a fibroblast-derived matrix deposited all over the scaffold.

After maturation, the generated dermis was inserted into the BoC and an FTSm was developed for 12 days. For this, HEKns were seeded on top of the dermis and, after 2 days under submerged conditions, medium was perfused via basal chamber. The scaffold structure allowed the passive diffusion of fresh nutrients to the basal layer of cells and waste products from the cells. Nutrient-depleted medium was pumped out of the chamber via the outlet. The apical surface of FTSm was exposed to ambient air to stimulate differentiation into a stratified layer at the ALI.

FTSm structure was monitored over time and histological cross-sections were analysed employing H&E staining to demonstrate the capability of FTSm to reflect the human skin anatomy (Figure 5). In the early culture phase, HEKns formed a continuous layer on top of the dermis (Figure 5 top right). After 4 to 6 days of culture at the ALI, cells in the basal layer began to develop a cubic morphology and HEKns appeared to flatten in higher cell layers. Furthermore, a single cell layer with granula and *stratum corneum* was detectable (Figure 5 middle). In the late culture phase, from 9 to 12 days, the epidermal differentiation was complete and the thickness of the viable cell layers and the thickness of the *stratum corneum* increased (Figure 5 bottom). After 12 days of culture, FTSm displayed a native human skin-like morphology, including a well-organized, orthokeratinized epidermis over a fibroblast-populated dermis (Figure 5 bottom right). H&E staining reveals all layers of the epidermis: *stratum basale*, *stratum spinosum*, *stratum granulosum* and *stratum corneum*. Other histological features include desmosome junctions between keratinocytes and it is possible to distinguish keratohyalin granules in the granular layer.

### 3.4. Real Time Measurement of Skin Barrier Formation

The integrity of the tissue barrier was evaluated by measuring TEER on-chip during 14 days of culture (Figure 6). To measure the TEER of these cultures, ALI was interrupted and PBS was pumped into the apical chamber. This completed the electrical circuit between the electrodes integrated in the apical chamber and the electrodes integrated in the basal chamber. The TEER of the tissue was recorded continuously for 10 min to capture fluctuations in the measurement. PBS was then removed from the apical chamber and air was pumped to maintain the ALI.

Under submerged conditions, the TEER values varied between 66 and 106 Ω·cm^2^ for the different tissue batches. During epidermal differentiation at ALI, electric resistance increased significantly, reaching 503 ± 150 Ω·cm^2^ after 4 days of culture at ALI and 663 ± 133 Ω·cm^2^ after 8 days of culture. At day 12, a mean value of 1050 ± 180 Ω·cm^2^ was obtained, indicating the formation of a permeability barrier and multi-layered epithelia. For all the independent batches of FTSm, the TEER values remained approximately stable after 12 days at ALI. The shift of the electrical properties of the FTSm during the epidermal differentiation could be attributed to architectural changes of the tissue with the progressive formation of tight-junctions and *stratum corneum*. The integrated home-made electrode showed stable TEER monitoring, without significant variation during the 10 min measurements.

### 3.5. Real Time TEER Measurement as a Parameter to Assess Drug Irritation

Figure 7a depicts the impact of chemical disruption of FTSm by exposing the upper layer to 0.2% SDS. TEER values were monitored continuously for 3 h and registered every 15 min. The batches of FTSm had mean TEER values of 1086 ± 239 Ω·cm^2^ before exposure. Upon exposure of the tissue to the test substance SDS at *t* = 0, the TEER values remained relatively stable for 15 min. After 0.25 h, the TEER value abruptly drops linearly, reaching a final value of 53 ± 66 Ω·cm^2^ at *t* = 3 h. The designed system allows for a continuous TEER observation during permeation studies, whereas static experiments only permit singular measurements before and after the experiment.

In parallel to the TEER measurements, morphological analysis of the SDS-treated and control FTSm was undertaken. The tissue damage induced by the cytotoxic benchmark and the resultant disruption of the skin barrier was visualized by topical application of 0.2% SDS for 0 (control), 1 and 3 h followed by the observation of the passive diffusion of the hydrophilic fluorescent dye, lucifer yellow (Figure 7b left). When the dye was applied to an untreated skin model, it was retained in the *stratum corneum* and no diffusion was observed in the epidermis or into the dermal layer, showing the integrity of the skin barrier for this molecule. For skin models treated with SDS, a total penetration of the dye could be seen (Figure 7b right). A complete lack of epidermal integrity was observed for skin models treated with 0.2% SDS for 3 h and it was possible to observe tissue damage.

## 4. Discussion

A BoC device consists in two perpendicular chambers separated by a membrane, establishing two separate compartments (basal and apical). Here, we improve on the conventional BoC setup by designing a flexible platform with a modular architecture and incorporated sensors. Contrary to conventional systems, with the developed BoC it is possible to integrate scaffolds which are typically an order of magnitude thicker than membranes. The developed BoC does not require plasma bonding and it is cleanroom-free.

The monitoring of tissue function inside the OoC devices was challenging, frequently requiring removal of tissue from its original housing. Conventional analysis occurred exclusively at the endpoint, limiting the possibility of monitoring changes over time. In this work, we integrated low-cost commercial electrodes in the BoC device for real-time TEER measurement during tissue formation. To reduce the measurement errors by variations in electrode placement, we integrated immobilized TEER electrodes directly within the chip model and in close proximity to the tissue. This not only reduces the contribution of electrical resistance from the cell culture medium, but also the signal noise generated by the electrode motion.

Odijik et al. pointed out the problems resulting from integrating electrodes in the conventional BoC configuration which result in a frequent overestimation of TEER values [26]. The group highlighted the importance of studying the geometry and placement of the electrodes to prevent measurement errors.

In our work, the sensitivity field along the tissue barrier area was evaluated using FEA for different tetrapolar electrode configurations and chamber geometries. The relevance of using FEA to study the sensitivity of tetrapolar electrode systems was experimentally confirmed by several studies using phantoms [36,37,38]. Overall, the simulations show that the sensitivity is dependent on the electrode geometry, cell culture area, TEER value and chamber geometry. In particular, the use of chopstick electrodes resulted in a non-uniform current distribution along the cell culture surface and a non-uniform sensitivity distribution. This resulted in an error when normalizing by multiplying by the culture area and a consequent overestimation of the TEER. Experiments using the calibration chamber showed that TEER readings with handheld chopstick electrodes are highly dependent on the electrode positioning and the variances in depth and angle of immersion result in low precision. These factors result in measurement errors, especially for larger culture inserts and can explain why, for the same cell and tissue types, a wide range of TEER values have been reported in the literature. These findings are in accordance with previous studies [15,26].

When using custom designed and embedded TEER electrodes, it is particularly important to ensure a uniform sensitivity across the tissue. To study the impact of the size electrodes integrated in the BoC, we compared a larger electrode assembly (covering ~ 65% of total culture area) to a smaller electrode assembly (covering ~ 15% of total culture area). No advantages in sensitivity distribution were seen using the larger electrodes in the BoC with the chosen geometry for TEERs ≥ 10^2^ Ω·cm^2^. Taking this into account, the smaller electrode configuration was chosen for BoC fabrication. For TEER values ≥10^3^ Ω·cm^2^, a homogenous sensitivity distribution was achieved with a maximum deviation of 10% for a symmetric tissue placement between the electrodes and 20% for asymmetric tissue placement (experimental conditions).

Various studies where electrodes are integrated on BoC devices highlight the importance of designing configurations compatible with the optical visualization of the cell culture. This is usually achieved by using a complex manufacturing process [21,39]. Although optical visualization can be useful for monitoring 2D cell cultures, its relevance for in vivo-like 3D tissues grown on membranes or scaffolds is very limited. In this study we integrated TEER electrodes that cover part of the cell culture area. Although this affects chamber visualization it is still possible to detect trapped bubbles, contaminations and other relevant findings. Furthermore, the modular chip architecture using magnets/screws made it possible to open the device and remove the cell culture insert for further testing and imaging techniques.

The designed BoC system was successfully fabricated and the custom-built electrodes based on the FEA were integrated on the system without disrupting the culture conditions. To test the potential of the fabricated BoC system to develop complex biological barrier tissues, a state-of-the-art FTSm was grown and monitored on-chip. ALI was successfully established on the BoC device, a key feature for epidermal differentiation. This feature is also ideal for establishing other complex models such as lung-on-a-chip since respiratory tract epithelial cells interact with both liquid and air in vivo [40].

To date, much progress has been made to develop biomimetic FTSm using OoC devices. However, none combine a functional OoC for 3D skin production with real-time TEER measurement. Ramadan and Ting performed TEER measurements on-chip to study interactions between HKns and monocytes; however, their system was only used for 2D cell culture with perfusion [41]. Alexander et al. modified an automated intelligent mobile lab for in vitro diagnostics (IMOLA-IVD) with integrated TEER sensors from cellasys to suit a reconstructed epidermis model [42]. However, the skin model was not developed on-chip. Our device could provide a reliable, low-cost platform for the formation of complex biological barriers such as FTSms and for the real-time, non-destructive monitoring of its function.

TEER analysis was used to evaluate the stage of skin differentiation and the impact of a topical irritant. Polarized keratinocytes began a complex differentiation process from day 0 under submerged conditions until day 12 at ALI, when a totally differentiated structure was already present. The increasing TEER levels with time during FTSm formation reflect the overall ion barrier of the stratum corneum and tight junction formation which regulate the movement of ions across the paracellular pathway. The FTSm reached values of 1050 ± 180 Ω·cm^2^, which are comparable to the values obtained in previous studies for similar models [43,44,45].

In addition to monitoring epidermal differentiation, TEER measurements have the potential to be employed as a real-time non-destructive analysis tool in the context of drug testing. In fact, TEER analysis is now used as a testing parameter of the Organisation for Economic Co-operation and Development (OECD) test guidelines (TG) 430 in order to classify ingredients for their capacity to modify the skin barrier (skin corrosivity in rat skin) [46]. To test the applicability of the developed device as a complement for skin toxicity testing, we performed TEER measurements while exposing the skin to a benchmark irritant (SDS) for 3 h. The disruption of barrier function was monitored in real-time and resulted in a decrease in TEER values. The impact of the irritant on the skin barrier was also shown by the penetration of the aqueous-soluble lucifer yellow dye. These experiments show the potential of the developed BoC with integrated sensors for on-chip assessment of tissue barrier integrity in the context of drug testing. In contrast to conventional testing procedures, TEER measurements allows real-time monitoring, comparison of individual tissues before and after treatment and can be combined with other standard endpoint assays.

## 5. Conclusions

In this work, we presented a low-cost BoC with a modular architecture and integrated electrodes for real-time, non-destructive TEER measurements. The integrated tetrapolar electrode configuration resulted in a more homogenous sensitivity distribution along the culture area when compared with conventional chopstick systems. Furthermore, its positioning reduced the contribution of the electrical resistance from the cell culture medium and the signal noise generated by the electrode motion. A state-of-the-art FTSm was developed inside the BoC by providing a dynamic microenvironment and by correctly establishing an ALI. TEER measurements were performed in real-time to monitor the tissue development and its increase reflected tight junction formation and *stratum corneum* accumulation. Further, our results suggest that the BoC platform incorporating embedded TEER sensors presents the opportunity of conducting drug testing directly on the platform, with key diagnostic parameters being monitored in real-time. In particular, TEER measurements can be useful to monitor the effect of irritants on biological barriers. Given the flexibility of the platform, additional cell types could be added to the FTSm and other epithelial tissues could be reconstructed and monitored. This advanced BoC system modelling complex biological barrier tissues (physiological and pathological states) presents the potential to revolutionize pharmaceutical, toxicological and cosmetic applications.

## Figures and Tables

**Figure 1 micromachines-12-00816-f001:**
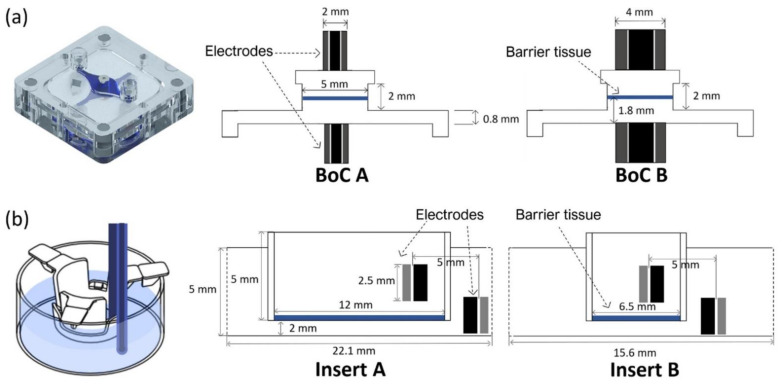
Geometrical dimensions for the simulated models. (**a**) BoC (barrier-on-chip) model with integrated concentric electrodes of different sizes: current carrying (CC) electrodes 0.8 mm + voltage sensing (VS) electrodes 2 mm diameter (BoC A) and CC electrodes 2 mm + VS electrodes 4 mm diameter (BoC B). The image on the left are the 3D representations of the experimental setup. (**b**) Insert model with chopstick electrodes and insert of different diameters: 12 mm diameter (Insert A) and 6.5 mm diameter (Insert B). The image on the left is a schematic representation of the measurements using chopstick electrodes and an insert. The barrier tissue is represented in blue.

**Figure 2 micromachines-12-00816-f002:**
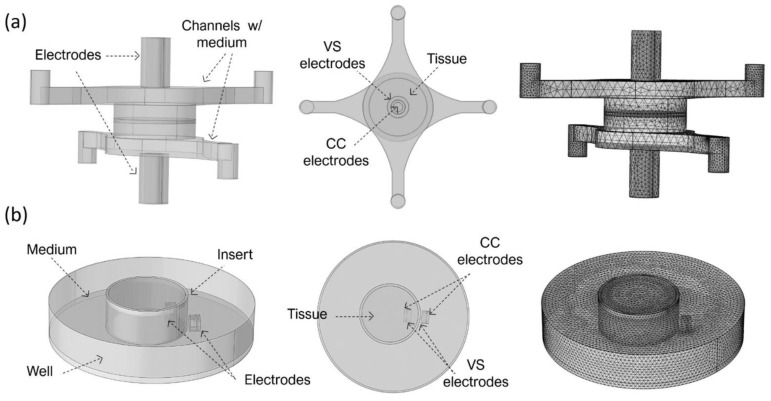
Three-dimensional geometry and created mesh for finite element analysis (FEA) for (**a**) the BoC model with integrated electrodes (CC 0.8 mm + VS 2 mm diameter) and tetrahedral mesh consisting of approximately 7 × 10^4^ domain elements, 2 × 10^4^ boundary elements, and 1 × 10^3^ edge elements and (**b**) the Insert model (insert of 12 mm diameter). Complete mesh consists of approximately 2 × 10^5^ domain elements, 2 × 10^4^ boundary elements, and 2 × 10^3^ edge elements.

**Figure 3 micromachines-12-00816-f003:**
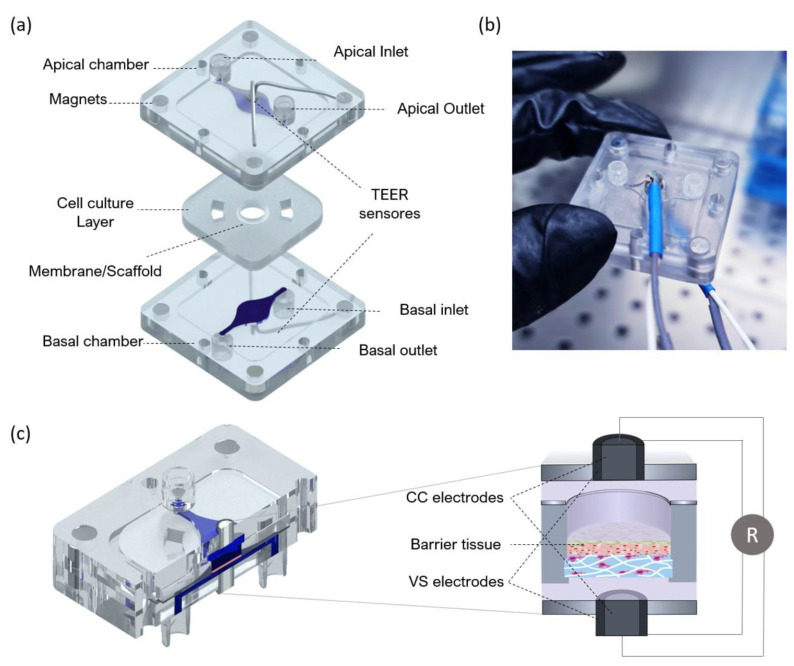
Design of the BoC with integrated electrodes for TEER measurements. (**a**) Schematic exploded view of the BoC platform. The device consists of central insert for cell culture and two external compartments (apical and basal) for top and bottom perfusion with air and/or medium and integrated electrodes. The central insert layer is composed of polydimethylsiloxane (PDMS) and a thin layer of polymerase chain reaction (PCR) tape sealing a membrane/scaffold. The top and bottom compartments are made of polystyrene and include magnets at the corners to correctly seal the insert layer and establish interfaces (air–liquid and liquid–liquid). (**b**) Picture of the assembled device with integrated TEER sensors made of Ag/AgCl-sintered pellet electrode, insulating polyether ether ketone (PEEK) tubing and external silver tubing. (**c**) Schematic representation of the BoC side view showing the fluid pathway, barrier tissue at the centre and concentric electrodes (CC and VS electrodes) positioned at the top and bottom chambers. The zoom-in panel showing the cross-section is not drawn to scale.

**Figure 4 micromachines-12-00816-f004:**
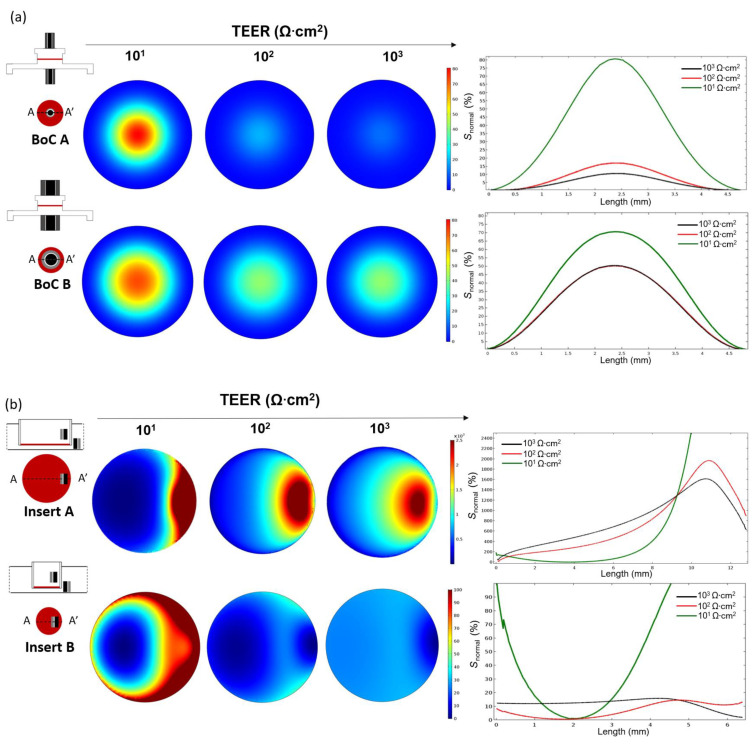
Normalized sensitivity distribution (*S*_normal_), including the planar sensitivity distribution along the cell barrier surface (rainbow colour map) in the x–y plane and sensitivity distribution along the cell barrier though the axis (lines AA’) shown in the scheme of the model at the left, when TEER was measured for (**a**) BoC device with concentric electrodes occupying 15% total area (BoC A) and occupying 65% total area (BoC B) and when TEER was measured (**b**) using chopstick electrodes in an insert culture insert with 12 mm diameter (Insert A) and 6.5 mm diameter (Insert B). Results are presented for different TEERS (10^1^–10^3^ Ω·cm^2^). An *S*_normal_ value of 0% represents the zone of lower sensitivity. Note that axes have different scales.

**Figure 5 micromachines-12-00816-f005:**
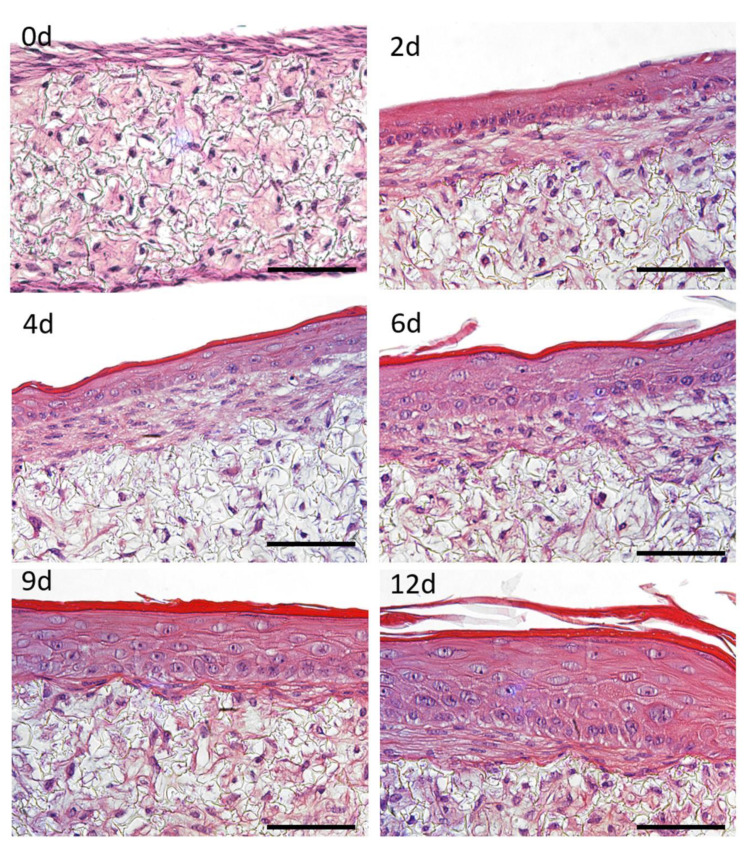
Representative images of haematoxylin and eosin (H&E)-stained histological cross sections of a full thickness skin model (FTSm) cultured for different times (0, 2, 4, 6, 9 and 12 days) at the air–liquid interface (ALI). Primary human epidermal keratinocytes isolated from neonatal foreskin (HEKns) were seeded on top of a dermis cultured for 12 days and maintained at ALI for up to 12 days to promote epidermal differentiation. Scale bars: 100 μm.

**Figure 6 micromachines-12-00816-f006:**
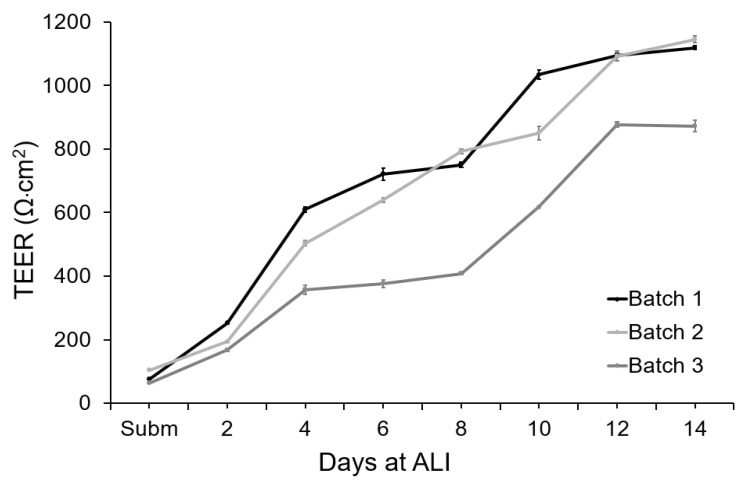
TEER measurements performed with the concentric electrodes integrated in the BoC device and an EVOM system. Measurements were performed from day 0 (submerged conditions) to day 14 at ALI for 3 different batches of FTSm grown on-chip (*N* = 3 and *n* = 1). Data represent the average TEER values registered during a 10 min measurement for each point. Data dispersion during this period was used to calculate the standard deviation. Error bars represent mean  ±  standard deviation.

**Figure 7 micromachines-12-00816-f007:**
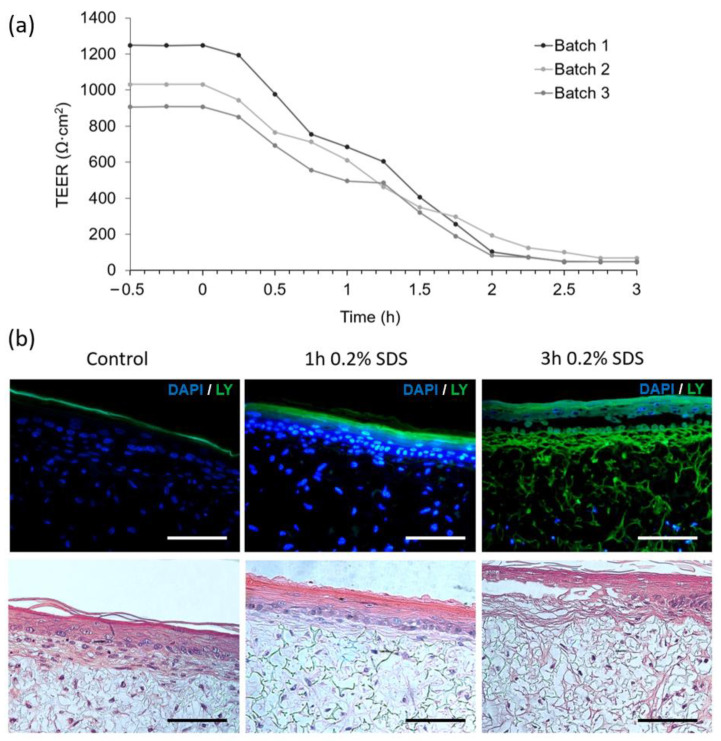
Assessment of barrier damage from the exposure to a benchmark irritant (0.2% SDS). (**a**) TEER measurements performed with concentric electrodes integrated in the BoC device and an EVOM system. Measurements were performed over time before and after exposure to 0.2% SDS in PBS (at time = 0) for 3 batches of FTSm grown for 14 days on chip. (**b**) Assessment of barrier resistance in FTSm: representative fluorescence (top) and H&E images (bottom) cultured for 14 days at ALI and exposed to 0.2% SDS for 0 (control), 1 and 3 h, followed by the application of the green fluorescent dye lucifer yellow (LY, green) for 30 min to assess barrier integrity. Cell nuclei were stained by DAPI (blue). Scale bars: 100 μm.

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
