# Peer review of "Barrier-on-a-Chip with a Modular Architecture and Integrated Sensors for Real-Time Measurement of Biological Barrier Function"

_micromachines, 2021, doi:10.3390/mi12070816_

Round 1
Reviewer 1 Report
Reviewer’s comment:
The authors describe an in vitro skin model based on a microfluidic device with integrated electrodes for TEER measurements. They estimated the sensitivity of the electrodes by Finite element analysis and fabricated the device. The device was used for the demonstration of TEER measurement of the human skin model to evaluate the function of the device. The results showed that the human skin model was established and TEER of the model was measured well on the chip. In addition, they performed the toxicity test using the device with the skin model as cell-based assays. The concept itself is not quite new, however an interesting description of the microfluidic platform and its potential applications have been given. I suggest publication of this paper but only after minor revisions per the following;
- Transwell is the product name for Corning's culture insert. It is inappropriate to use it as the name of a general cell culture insert. The shape of the culture insert in the picture in Fig. 1(b) is clearly not a Transwell, although it is not actually used in the experiments in this paper.
- In Fig. 4, the values and ranges of the Y-axis should be aligned to compare between graphs. Also, in TranswellA, the digits of the Y-axis values are different, is this correct?
- The meaning of using a calibration chamber was unclear. Please add a detailed explanation in the manuscript.
- I agree that the uniformity of sensitivity distribution is important, however I think the absolute value of sensitivity of a sensor is also important. In the paper, the authors discuss normalized sensitivity. However, the authors also need to discuss the absolute value of sensitivity of TEER measurements for monitoring the conditions of cells and tissues.
- The unit of the TEER value in Fig. 6 and Fig. 7 is kΩ. Usually, the unit of TEER value is Ω-cm^2. The units of the TEER values should be the same to make it easier to compare with values in other references.
Author Response
We thank the reviewers for their time and insightful feedback. We have incorporated various suggestions made by the reviewers, added more information in the introduction and methods sections. We also added supplemental information. Bellow we address all the reviewer’s comments and concerns (the line numbers were counted without track changes activated):
Reviewer 1
Transwell is the product name for Corning's culture insert. It is inappropriate to use it as the name of a general cell culture insert. The shape of the culture insert in the picture in Fig. 1(b) is clearly not a Transwell, although it is not actually used in the experiments in this paper.
Author Response: Thank you for pointing out that imprecision! We were using the name Transwell for the concept of a permeable cell culture insert but it is was incorrect. We have substituted the word “Transwell” for “cell culture insert” or “insert” throughout the manuscript. We chose to model inserts with conventional sizes (6.5 mm and 12 mm diameter) which are available from Corning (Transwel) but are also available from other manufacturers. These diameters are widely used to generate biological barriers and are recommended for normalizing TEER measurements performed with chopstick electrodes.
In Fig. 4, the values and ranges of the Y-axis should be aligned to compare between graphs. Also, in TranswellA, the digits of the Y-axis values are different, is this correct?
Author Response: Thank you for pointing out. Although the graphs for BoC A and BoC B have data in the same ranges, the maximum values in the Y-axis were slightly different. We have now corrected this problem. Regarding the comparison between the BoC systems and transwell (insert) system, the different electrode configurations exhibited different ranges of sensitivity values by up to two orders of magnitude. The high values of sensitivity for transwellA (insertA) can be interpreted as the resistance measurements performed with this configuration being only representative of a small zone of the total culture area. Taking into account the difference in range by orders of magnitude, the range had to be set individually for each plot to preserve important information.
The meaning of using a calibration chamber was unclear. Please add a detailed explanation in the manuscript.
Author Response: We added more information regarding the calibration chamber to the manuscript (Methods and materials sections: line 320-328, without track changes). The calibration chamber was fabricated to allow direct comparison between the electrodes integrated in the BoC and the conventional chopstick electrodes. Although chopstick electrodes present several limitations, many of them described in the present manuscript, they are considered the gold standard for TEER measurements in small inserts (≤6.5 mm diameter), and present approximate homogeneous current distributions. Since it is not possible to directly introduce the chopstick electrodes on the BoC device, a custom calibration chamber was designed and fabricated using precision milling. The fabricated chamber could accommodate the cell culture layer from the BoC and simultaneously included openings where the conventional chopstick electrodes could be introduced.
I agree that the uniformity of sensitivity distribution is important, however I think the absolute value of sensitivity of a sensor is also important. In the paper, the authors discuss normalized sensitivity. However, the authors also need to discuss the absolute value of sensitivity of TEER measurements for monitoring the conditions of cells and tissues.
Author Response: We agree that it is relevant to show the absolute values of sensitivity. We added to the supplementary materials, the sensitivity map (absolute values) along the cell culture area and a sensitivity map along a longitudinal cut, using BoC A (Figure S2). The unit of sensitivity is 1/m4. It is important to note that this concept of sensitivity is different from what is commonly used for sensors. Here, the sensitivity field of a tetrapolar system is a measure of how much a small volume contributes to the total measured resistance. The importance of studying the sensitivity distribution of the tetrapolar electrodes was first described by Grimnes et al [1]. The group discovered that these electrodes are prone to errors by producing a non-homogenous distribution of sensitivity, including zones of negative sensitivity. This is counterintuitive in nature and means that increased resistivity in that volume results in a lower total measured resistance. This phenomenon can be seen in the added supplementary Figure S2, where zones of negative sensitivity can be seen next to the electrodes.
Regarding the concept of sensitivity commonly described as the ratio between the output signal and measured property and, in this particular case, the sensitivity of TEER electrodes for monitoring barrier function, it is very difficult to discuss absolute values. From figure 6 and 7 it is possible to see that the device can capture the formation of a biological barrier (skin barrier) as well as its destruction when exposing to a benchmark irritant. This can be particularly understood when comparing with the histologic images and lucifer yellow die penetration. In the future, these studies could be complemented by performing a more subtle damage (for e.g. perfusing the tissue with cytokines such as TNF-alfa) and comparing the TEER measurements with permeability assays. This will be the focus of a future paper focused on generating an inflamed skin model on the platform.
The unit of the TEER value in Fig. 6 and Fig. 7 is kΩ. Usually, the unit of TEER value is Ω-cm^2. The units of the TEER values should be the same to make it easier to compare with values in other references.
Author Response: We changed the graph and now the values use ohm*cm^2 to make it easier to compare with the references. This alteration can be done since, from the simulations, it seems that the sensitivity is considerable homogeneous, meaning that all the cell culture area contributes to the measured resistance. This cannot be said for other configurations or when combining chopstick electrodes with larger inserts. On these systems, the resistance measurements are only representative of a small zone of the total culture area, revealing an error when multiplying the measured values by the total area.
References:
- Grimnes S, Martinsen Ø. Sources of error in tetrapolar impedance measurements on biomaterials and other ionic conductors. J Phys D: Appl Phys. 2006;40:9-14. doi:10.1088/0022-3727/40/1/S02

Reviewer 2 Report
The manuscript describes the barrier-on-a-chip to monitor the skin's biological barrier formation by trans-epithelial electrical resistance (TEER). The authors presented simulated results of TEER sensitivities and formation of skin tissues by utilizing air-liquid interface (ALI), and the skin damage model.
Unfortunately, this manuscript was not well-organized, and hard to follow their logic. There are other issues to be addressed as written below.
<Major points>
- In the introduction, the authors described the issues to integrate TEER electrodes in organs-on-a-chip (OoC) to ensure that the electrical current density would be uniform across the cell layer by line 74. However, from line 75, the authors then mentioned "low cost" and "modular architecture" for their OoC, which had never been discussed. The authors also described "cleanroom-free" fabrication on line 77, but never discussed earlier. Thus, the authors' logic flow to introduce the background of research and their objective in this manuscript was not organized well. I strongly recommend reorganizing the logic in the introduction intensively.
- On line 88, the authors mentioned "none combines a functional OoC for 3D skin production with real-time TEER measurement." But why? Is there any technical difficulty? Is there no need to combine? The authors need to clarify.
- Although the authors showed simulation results of TEER sensitivities with different TEER settings in Fig. 4, they did not validate these simulations by using real TEER experiments. I think this is the most critical experiment to be carried out for this manuscript. I strongly suggest fabricating both BoC A and BoC B, and analyzing the sensitivity differences between them.
- The fabrication process provided in the manuscript was insufficient to be reproduced by the readers. For example, the thickness of PDMS membrane was not provided. How is it fabricated? For PDMS curing, which temperature was used?
- Figs. 6 and 7(a) show TEER values, but their Y-axis showed only ohm. In general, TEER values were used ohm*cm^2. The authors need to address this. Moreover, the authors showed error bars, but did not mention how many experiments were repeated.
- Fig. 7(b)'s legend did not show which staining was shown in green or blue.
- The authors used SDS to make the cell barrier damaged. The leakage of lucifer yellow is one of way to visualize, but the authors should show the disruption of tight junction by immunostaining for ZO-1.
Author Response
We thank the reviewers for their time and insightful feedback. We have incorporated various suggestions made by the reviewers, added more information in the introduction and methods sections. We also added supplemental information. Bellow we address all the reviewer’s comments and concerns (the line numbers were counted without track changes activated):
Reviewer 2
In the introduction, the authors described the issues to integrate TEER electrodes in organs-on-a-chip (OoC) to ensure that the electrical current density would be uniform across the cell layer by line 74. However, from line 75, the authors then mentioned "low cost" and "modular architecture" for their OoC, which had never been discussed. The authors also described "cleanroom-free" fabrication on line 77, but never discussed earlier. Thus, the authors' logic flow to introduce the background of research and their objective in this manuscript was not organized well. I strongly recommend reorganizing the logic in the introduction intensively.
Author Response: Thank you for pointing out the organization problems in this manuscript. We agree that the introduction was missing key sentences connecting the existent problems and the proposed solutions. We added more information to the introduction which, hopefully, will create a better logic flow. More precisely, we added information about existent solutions for Organ-on-a-chip (OoC ) with integrated electrodes and what we consider to be some of their main limitations:
“…The integration of the electrodes during the fabrication reduces the noise generated by electrode motion. However, most of the devices described in the literature require expensive processes and advanced microfabrication techniques for the development of the OoC and for the patterning of the TEER electrodes on the substrates of the channels, often requiring cleanroom access. Furthermore, the process of PDMS bonding makes it difficult to integrate scaffolds for 3D cell culture which are typically an order of magnitude thicker than membranes.”
And present our proposed solution:
“Here, we have developed a low-cost chip with a modular architecture and integrated custom-made tetrapolar electrodes for development and monitoring of a biological barrier-on-chip (BoC). The fabrication of this platform does not require plasma bonding and it is cleanroom-free. The geometry of the custom-made electrodes integrated on the BoC is similar to the electrode arrangement on the EndOhm chambers, with a symmetric and concentric placement on both sides of the cellular barrier. Taking into account works from Odijik et al and Yeste et al highlighting possible sources of error when integrating tetrapolar electrodes on OoC, we use finite element analysis (FEA) to determine the sensitivity distribution within the cell culture chamber using the custom-made electrodes...”
We performed various changes throughout the introduction specially from line 60 to 98 (line numbers counted without tack changes activated).
On line 88, the authors mentioned "none combines a functional OoC for 3D skin production with real-time TEER measurement." But why? Is there any technical difficulty? Is there no need to combine? The authors need to clarify.
Author Response: Skin-on-a-chip is a relatively new field of research with the first paper mentioning the topic being published in 2013. The generation of a full thickness skin (dermis and epidermis) inside an OoC presents several technical difficulties (Introduction: line 93-99).
“Various skin-on-a-chip (SoC) devices have been fabricated in an attempt of reproducing a skin model with a better barrier function by including fluidic forces. However, SoC devices face multiple technical challenges including the tendency of the hydrogel-based materials (e.g., collagen) to contract over time leading to tissue disruption and the difficulty in recreating the complex 3D architecture of the epidermis. To our knowledge, until now, there is no description of a functional OoC combining 3D skin production with real-time TEER measurement”
One of the most relevant technical difficulties is the contraction of the dermis with time which would make TEER measurements and permeation testing impossible. On example of OoC suffering from the collagen contraction can be seen here [2]. In the last years, our group has been perfecting a protocol to produce a full thickness skin model which does not include animal components and has the ideal stability to be used in the context of organ-on-a-chip. This protocol was published recently [3]. Other groups are doing the same: For example, to avoid the limitations of the collagen contraction, Sriram G. et al developed a full-thickness SOC using a fibrin-based dermal matrix and a thermoplastic-based chip [4]. The group was able to successfully produce full thickness skin. However, they did not include sensors in the platform and only introduced handheld sensors in the inlets/outlets as an endpoint assay, meaning it would not be possible, for example, to study skin irritation in real-time with their platform. On the other hand, Alexander et al used an automated intelligent mobile lab for in vitro diagnostics (IMOLA-IVD) with integrated TEER sensors from Cellasys but did not show the potential of their chip to generate the tissues [5] (Discussion: Line 610-619).
Although the authors showed simulation results of TEER sensitivities with different TEER settings in Fig. 4, they did not validate these simulations by using real TEER experiments. I think this is the most critical experiment to be carried out for this manuscript. I strongly suggest fabricating both BoC A and BoC B, and analyzing the sensitivity differences between them.
Author Response: We agree that it would be interesting to experimentally validate the results obtained from the simulations. However, there are several limitations and other considerations that need to be addressed: It is not possible to experimentally replicate the results from the simulations, namely, the sensitivity distribution along the cell culture volume, since the current density vectors at a specific point within the volume cannot be directly measured. By using the reciprocity and lead field theory in conjunction with finite element modelling the distribution of the sensitivity field can be calculated. Grimes et al proposed this model as an important tool to easily plot the sensitivity fields of a measuring setup, thereby avoiding important errors that arise from the use of tetrapolar systems (zones of negative sensitivity, zones of zero sensitivity and non-homogeneity) [1]. Since then, many groups used these simulations to design their tetrapolar setups specially in the context of bioimpedance.
We agree that is possible to roughly validate the obtained results by performing phantom measurements, however, this was already performed multiple times in the literature [6-8] (Discussion: line 569-571). In the phantom experiments, plastic cylinders are placed in a volume conductor with the electrodes fixed on the inside wall. The change of resistance is calculated for each cylindrical object position within the fantom and the fractional change with respect to the background is defined as the sensitivity. However, it is important to note that these phantom experiments are performed in the context of bioimpedance to probe organs in the human body and the volumes available to perform them are orders of magnitude higher than the ones used in the context of organ-on-a-chip. In our particular case, it would be an enormous technical challenge to perform phantom experiments inside the available volumes and, since we are using models than were already applied and tested in the past, we don’t believe it is necessary.
The fabrication process provided in the manuscript was insufficient to be reproduced by the readers. For example, the thickness of PDMS membrane was not provided. How is it fabricated? For PDMS curing, which temperature was used?
Author Response: Yes, we agree that important information was missing and it was not possible to reproduce our design. We added more information to the Methods section regarding the mold design and fabrication, necessary to produce the PDMS layer, as well as more information for PDMS curing (Methods: Line 204 – 237).
Figs. 6 and 7(a) show TEER values, but their Y-axis showed only ohm. In general, TEER values were used ohm*cm^2. The authors need to address this. Moreover, the authors showed error bars, but did not mention how many experiments were repeated.
Author Response: We changed the graph and now the values use ohm*cm^2 to make it easier to compare with the references. This alteration can be done since, from the simulations, it seems that the sensitivity is considerable homogeneous, meaning that all the cell culture area contributes to the measured resistance. This cannot be said for other configurations or when combining chopstick electrodes with larger insets. On these systems, the resistance measurements are only representative of a small zone of the total culture area, revealing an error when multiplying the measured values by the total area.
TEER measurements were performed during the formation of 3 biological independent repetitions (N=3). One technical repetition was performed (n=1). Here, the standard deviation represents the dispersion during a 10-minute measurement window to reflect the electrode performance. For example, using chopstick electrodes during the 10-minute window the value oscillates due to variations in the angle and depth of immersion and deviates between 15-37%. The small error bar shows the stability of the sensors during the measurement time window. We added more information to the legend.
Fig. 7(b)'s legend did not show which staining was shown in green or blue.
Author Response Thank you for pointing this out. We now added this information to the Figure 7b and its legend.
The authors used SDS to make the cell barrier damaged. The leakage of lucifer yellow is one of way to visualize, but the authors should show the disruption of tight junction by immunostaining for ZO-1.
Author Response Thank you for this suggestion! It would be interesting to study the disruption of tight junction by immunostaining for ZO-1. However, for the particular case where the skin is damaged with SDS, we believe that not much information would be gained. Since the damage is so extreme, the tissue disruption can be clearly seen not only though the leakage of lucifer yellow but also through histological images which shows extreme damage. We believe that the proposed technique would be very useful to study a more subtle damage, for example, the impact of TNF-alfa on thigh junctions. Our group is interested in evaluating this phenomenon and are interested in adding these results to our following publication focused on a detailed characterization of the skin produced on the chip. The current publication is intended as an introduction to the developed platform with integrated sensors, which we believe could be useful for researchers wanting a low-cost alternative to conventional organ-on-a-chip devices.
References:
- Grimnes S, Martinsen Ø. Sources of error in tetrapolar impedance measurements on biomaterials and other ionic conductors. J Phys D: Appl Phys. 2006;40:9-14. doi:10.1088/0022-3727/40/1/S02
- Mori N, Morimoto Y, Takeuchi S. Skin integrated with perfusable vascular channels on a chip. Biomaterials. 2017;116:48-56. doi:10.1016/j.biomaterials.2016.11.031
- Zoio P, Ventura S, Leite M, Oliva A. Pigmented full-thickness human skin model based on a fibroblast-derived matrix for long-term studies [published online ahead of print, 2021 Jun 19]. Tissue Eng Part C Methods. 2021;10.1089/ten.TEC.2021.0069. doi:10.1089/ten.TEC.2021.0069
- Sriram G, Alberti M, Dancik Y et al. Full-thickness human skin-on-chip with enhanced epidermal morphogenesis and barrier function. Materials Today. 2018;21(4):326-340. doi:10.1016/j.mattod.2017.11.002
- Alexander FA, Eggert S, Wiest J. Skin-on-a-Chip: Transepithelial Electrical Resistance and Extracellular Acidification Measurements through an Automated Air-Liquid Interface. Genes (Basel). 2018;9(2):114. doi:10.3390/genes9020114.
- Roy SK, Karal MAS, Kadir MA, Rabbani KS. A new six-electrode electrical impedance technique for probing deep organs in the human body. Eur Biophys J. 2019;48(8):711-719. doi:10.1007/s00249-019-01396-x
- Canali C, Mazzoni C, Larsen LB, et al. An impedance method for spatial sensing of 3D cell constructs--towards applications in tissue engineering. Analyst. 2015;140(17):6079-6088. doi:10.1039/c5an00987a.
- Rabbani KS. Focused Impedance Method: Basics and Applications. In: Simini F., Bertemes-Filho P, eds. Bioimpedance in Biomedical Applications and Research. Springer, Cham. 2018: 137-186. doi:10.1007/978-3-319-74388-2_9.
Reviewer 3 Report
“Barrier-on-a-chip with a modular architecture and integrated sensors for real-time measurement of biological barrier function”
Zoio et al. describe the development of a low-cost barrier-on-chip device with integrated electrodes to study the real-time monitoring of biological barriers. Using their system, they studied and monitor TEER changes of a full-thickness human skin model when stimulated with a benchmark irritant. Additionally, COMSOL based finite element modeling was performed to evaluate the sensitivity of these integrated electrodes.
The findings of this study suggest a new method for the development of barrier on chip and TEER monitoring. The implications of this study can help better monitoring during drug development and disease modeling
Major Weaknesses
- Readers would highly benefit from more detailed methods of chip sectioning used for histochemistry analysis.
- It would be great if the authors will show a more detailed characterization of skin model such as the expression of KRT-10, Flaggrin, Involucrin and KRT-16.
Author Response
We thank the reviewers for their time and insightful feedback. We have incorporated various suggestions made by the reviewers, added more information in the introduction and methods sections. We also added supplemental information. Bellow we address all the reviewer’s comments and concerns (the line numbers were counted without track changes activated):
Reviewer 3
Readers would highly benefit from more detailed methods of chip sectioning used for histochemistry analysis.
Author Response: Due to the modular architecture and reversible bond existent on the chip, the platform could be easily opened and the tissue removed. The histochemistry analysis was performed using standard methods, similar to the ones described in [1]. It was not necessary to employ any special method for histochemistry analysis. We added the used reference to the text.
It would be great if the authors will show a more detailed characterization of skin model such as the expression of KRT-10, Flaggrin, Involucrin and KRT-16.
Author Response: We agree with the suggestion. However, the goal of the present paper is to introduce the platform with integrated electrodes and the skin model is only used as a preliminary proof of concept. We agree on the importance of performing immunodetection of specific markers to better characterized the produced tissues and, in a paper published recently by our group, we performed a detailed characterization of the skin model created off chip using similar tissue engineering techniques [2]. Since a detailed characterization will require a detailed discussion, this will be performed on our next paper which will compare the skin produced inside the platform and a skin produced using conventional methods.
References:
- Capallere C, Plaza C, Meyrignac C, et al. Property characterization of reconstructed human epidermis equivalents, and performance as a skin irritation model. Toxicol In Vitro. 2018;53:45-56. doi:10.1016/j.tiv.2018.07.005
- Zoio P, Ventura S, Leite M, Oliva A. Pigmented full-thickness human skin model based on a fibroblast-derived matrix for long-term studies [published online ahead of print, 2021 Jun 19]. Tissue Eng Part C Methods. 2021;10.1089/ten.TEC.2021.0069. doi:10.1089/ten.TEC.2021.0069
Round 2
Reviewer 2 Report
The authors tried to revise their manuscript to fulfill the requirements from me. In some parts, the authors were able to address, but unfortunately, they could not address the other parts. I would like to suggest again for them.
1. In the last manuscript, I could not find why the authors and their chip need to have "low-cost" and "cleanroom-free" process to fabricate. In the revised manuscript, they mentioned, "However, most of the devices described in the literature require expensive processes and advanced microfabrication techniques for the development of the OoC and for the patterning of the TEER electrodes on the substrates of the channels, often requiring cleanroom access." and “Here, we have developed a low-cost chip with a modular architecture and integrated custom-made tetrapolar electrodes for development and monitoring of a biological barrier-on-chip (BoC). The fabrication of this platform does not require plasma bonding and it is cleanroom-free." But, again, I could not understand why they need to have "low-cost" and "cleanroom-free". In other words, why do costs and cleanrooms cause the issues for the chips? The authors should describe the current issues, and how they can address these issues.
2. Last time, I mentioned, "Although the authors showed simulation results of TEER sensitivities with different TEER settings in Fig. 4, they did not validate these simulations by using real TEER experiments. I think this is the most critical experiment to be carried out for this manuscript. I strongly suggest fabricating both BoC A and BoC B, and analyzing the sensitivity differences between them." The authors partially agreed, but did not agree to carry out the experiment, since these results have been published already in other papers. In that case, why do the authors need to show these results in the manuscript? I've known these simulated results were published, and thus suggested adding the new results to support the authors' manuscript. The authors mentioned that it is difficult to validate this simulation, because of sensitivity distribution. But they can. The authors could form their skin model at the half area of cell culture insert and see the TEER values with different electrode settings. Thus, the authors will be able to validate their simulation briefly.
Author Response
Dear reviewer,
Once more we want to thank you for your thorough reading of our manuscript and for your constructive comments and suggestions which help improve the quality of our work. Hopefully we can answer some of your questions and concerns:
In the last manuscript, I could not find why the authors and their chip need to have "low-cost" and "cleanroom-free" process to fabricate. In the revised manuscript, they mentioned, "However, most of the devices described in the literature require expensive processes and advanced microfabrication techniques for the development of the OoC and for the patterning of the TEER electrodes on the substrates of the channels, often requiring cleanroom access." and “Here, we have developed a low-cost chip with a modular architecture and integrated custom-made tetrapolar electrodes for development and monitoring of a biological barrier-on-chip (BoC). The fabrication of this platform does not require plasma bonding and it is cleanroom-free." But, again, I could not understand why they need to have "low-cost" and "cleanroom-free". In other words, why do costs and cleanrooms cause the issues for the chips? The authors should describe the current issues, and how they can address these issues.
Some of the major barriers delaying the transfer of organ-on-a-chip technology to the industry are the costs of manufacturing and experimental implementation [1]. Taking into account that the end goal of these platforms is to substitute/complement conventional cell culture models for the development of new drugs and treatments, these barriers need to be surpassed. The successful translation of organ-on-a-chip will require the use of low-cost materials and fabrication strategies suitable for mass production. The device proposed in the present manuscript has these necessary characteristics. Although this is not the only differentiating feature of the device, we believe it is an important one and it is addressing a relevant issue. We added more information to the introduction (Line 76) and one more reference [1].
Last time, I mentioned, "Although the authors showed simulation results of TEER sensitivities with different TEER settings in Fig. 4, they did not validate these simulations by using real TEER experiments. I think this is the most critical experiment to be carried out for this manuscript. I strongly suggest fabricating both BoC A and BoC B, and analyzing the sensitivity differences between them." The authors partially agreed, but did not agree to carry out the experiment, since these results have been published already in other papers. In that case, why do the authors need to show these results in the manuscript? I've known these simulated results were published, and thus suggested adding the new results to support the authors' manuscript. The authors mentioned that it is difficult to validate this simulation, because of sensitivity distribution. But they can. The authors could form their skin model at the half area of cell culture insert and see the TEER values with different electrode settings. Thus, the authors will be able to validate their simulation briefly.
The purpose of the simulation was to make sure the specific geometry would not result in zones of zero/negative sensitivity or high inhomogeneity in the cell culture area. As we mentioned, we believe the strategy/model using reciprocity and lead field theory in conjunction with finite element modelling was experimentally validated in the literature. There is evidence that the model correctly replicates the behaviour of a tetrapolar electrode system in regards to the sensitivity distribution. The strategy/model itself was validated but it needed to be applied to our specific electrode + chip geometry. This was performed during the design/conceptualization phase and guided our decisions before the fabrication phase. Regarding the experimental validation using our device, it is well documented that, for example, “a 99.6% cell coverage of the measured TEER value will be 80% lower than the TEER of a cell culture with full cell coverage” [2]. This was also verified by our group by measuring the resistance of ruptured/damaged skin tissues. Thus, unfortunately, growing a skin model at the half area of the cell culture insert would not help. The effect would be too extreme, and would result in no opposition to the current (almost zero resistance when subtracting the blank). To perform the experiments, we would have to grow multiple skin tissues on the platform and create some type of controlled damage at the micrometre scale, in specific coordinates. This controlled damage would have to be exactly the same for each coordinate. Furthermore, the considerable variability in TEER readings between different skin models would greatly affect data interpretation. We believe this complicated process would only be justified if we were one of the first groups using the strategy proposed by Grimnes et al [3]. In general, numerical approaches are recognized as reliable methods to reducing cost, time, and effort while being relatively accurate.
References:
- Ramadan Q, Zourob M. Organ-on-a-chip engineering: Toward bridging the gap between lab and industry. Biomicrofluidics. 2020;14(4):041501. doi:10.1063/5.001158
- Odijk M, van der Meer AD, Levner D, et al. Measuring direct current trans-epithelial electrical resistance in organ-on-a-chip microsystems. Lab Chip. 2015;15(3):745-752. doi:10.1039/c4lc01219d
- Grimnes S, Martinsen Ø. Sources of error in tetrapolar impedance measurements on biomaterials and other ionic conductors. J Phys D: Appl Phys. 2006;40:9-14. doi:10.1088/0022-3727/40/1/S02